# Norepinephrine Leads to More Cardiopulmonary Toxicities than Epinephrine by Catecholamine Overdose in Rats

**DOI:** 10.3390/toxics8030069

**Published:** 2020-09-16

**Authors:** Wen-Hsien Lu, Hsin-Hung Chen, Bo-Hau Chen, Jui-Chen Lee, Chi-Cheng Lai, Che-Hsing Li, Ching-Jiunn Tseng

**Affiliations:** 1Department of Pediatrics, Kaohsiung Veterans General Hospital, Kaohsiung 813, Taiwan; lu6802@gmail.com; 2School of Medicine, National Yang-Ming University, Taipei 112, Taiwan; 3Institute of Biomedical Sciences, National Sun Yat-sen University, Kaohsiung 804, Taiwan; 4Department of Medical Education and Research, Kaohsiung Veterans General Hospital, Kaohsiung 813, Taiwan; shchen0910@gmail.com (H.-H.C.); xu3bjo45p@gmail.com (J.-C.L.); 5Department of Pediatrics, Taoyuan Armed Forces General Hospital, Taoyuan 325, Taiwan; dreamvenice@hotmail.com; 6Department of Cardiology, Kaohsiung Municipal United Hospital, Kaohsiung 804, Taiwan; llccheng@gmail.com; 7School of Medicine, Chung Shan Medical University, Taichung 402, Taiwan; kevinlee075@gmail.com; 8Department of Medical Research, China Medical University Hospital, China Medical University, Taichung 404, Taiwan

**Keywords:** norepinephrine, epinephrine, catecholamine overdose, ventricular dysfunction, lung injuries

## Abstract

While catecholamines like epinephrine (E) and norepinephrine (NE) are commonly used in emergency medicine, limited studies have discussed the harm of exogenously induced catecholamine overdose. We investigated the possible toxic effects of excessive catecholamine administration on cardiopulmonary function and structure via continuous 6 h intravenous injection of E and/or NE in rats. Heart rate, echocardiography, and ventricular pressure were measured throughout administration. Cardiopulmonary structure was also assessed by examining heart and lung tissue. Consecutive catecholamine injections induced severe tachycardia. Echocardiography results showed NE caused worse dysfunction than E. Simultaneously, both E and NE led to higher expression of Troponin T and connexin43 in the whole ventricles, which increased further with E+NE administration. The NE and E+NE groups showed severe pulmonary edema while all catecholamine-administering groups demonstrated reduced expression of receptor for advanced glycation end products and increased connexin43 levels in lung tissue. The right ventricle was more vulnerable to catecholamine overdose than the left. Rats injected with NE had a lower survival rate than those injected with E within 6 h. Catecholamine overdose induces acute lung injuries and ventricular cardiomyopathy, and E+NE is associated with a more severe outcome. The similarities of the results between the NE and E+NE groups may indicate a predominant role of NE in determining the overall cardiopulmonary damage. The results provide important clinical insights into the pathogenesis of catecholamine storm.

## 1. Introduction

The autonomic nervous system regulates human responses to environmental change. When humans are subjected to threatening events, sympathetic neurons release catecholamines like epinephrine (E) and norepinephrine (NE) to engage the fight-or-flight response. Adrenergic receptors (AR) are involved in stress-induced physiological processes through binding with catecholamines to regulate vascular resistance and heart rate. These neurohormonal mechanisms are mediated from the sympathetic nervous system, such as baroreflex sensitivity for adrenergic signaling [1].

Catecholamines are not only important for circulation physiology but are also vital for several emergency situations, such as cardiopulmonary arrest and septic shock. The latest version of the advanced cardiac life support (ACLS) guidelines, focusing on cardiopulmonary resuscitation published by the American Heart Association in 2019, suggests up to 1 mg E to be administered intravenously (IV) every 3–5 min in the event of cardiac arrest [2]. E binds to β1 AR, stimulating myocardial contraction, enabling patients with cardiac arrest to maintain mean arterial pressure and regain pulse recovery [3]. NE mainly binds to α receptors to induce vasoconstriction and blood pressure elevation; however, reflex bradycardia occurs simultaneously and thus restricts the clinical use of NE in cardiac arrest [4]. On the other hand, NE is strongly recommended as the first-choice vasopressor in cases of sepsis and septic shock [5], with an average of 8–12 μg/min IV to be continuously infused.

Nevertheless, a dysregulated release of catecholamines may also lead to severe cardiac toxicities. Administration of high dose isoproterenol, which is a synthetic catecholamine and β-AR agonist, leads to severe myocardial injury in rats [6]. This adverse effect is induced by oxidative stress in the myocardium [7]. The most common disease associated with catecholamine dysregulation is pheochromocytoma which is a persistent catecholamine-secreting tumor originating from chromaffin cells. The pheochormocytoma has adrenergic, noradrenergic, or dopaminergic biochemical phenotypes. Noradrenergic phenotype is more likely to present with hypertension and tachycardia, thus leading to cardiac dysfunction, compared to others [8].

Moreover, destruction of nucleus tractus solitarii (NTS) in the brainstem, involved in the autonomic center in the central nervous system, may also contribute to catecholamine dysregulation. Previously, we found that there was an acute increase in catecholamines, including E and NE, in peripheral blood following the administration of 6-hydroxydopamine (6-OHDA) to induce brainstem lesions in rats, which then led to an acute elevation of blood pressure [9]. This caused cardiac dysfunction, myocardial damage, and pulmonary edema within 7 h. Moreover, during this process, α1 receptors played a more significant role than β1 receptors enabling prazosin, a selective α1 receptor antagonist, to facilitate preservation of cardiac output (CO) and reverse pulmonary hemorrhagic edema [10]. Diseases involving brainstem deterioration may present similar symptoms such as cardiopulmonary dysfunction. For instance, traumatic brain injuries may also induce systolic dysfunction [11]. Decompensation or death, due to respiratory distress, and as a consequence of pulmonary edema, was observed within 4 days due to impairment of NTS from brainstem encephalitis (i.e., severe enterovirus 71 infection) [12].

Although some studies have compared the use of standard and high doses of E in cardiac arrest patients, reporting no significant higher risk of complications [13,14]; there are limited studies reporting on the adverse effects induced by catecholamine overdose on the cardiopulmonary system. To understand whether very high dose catecholamine treatment is related to severe dysfunction and damage of cardiovascular and pulmonary systems, normal rats were administered E or NE or E+NE IV continuously to simulate and evaluate an experimental model, testing the effects of external excessive catecholamine administration, on cardiopulmonary pathological and functional changes. We hypothesized that catecholamine overdose would lead to severe cardiopulmonary impairment. The difference in sensitivity to catecholamine-induced injuries between the right and left ventricles (RV and LV) was also evaluated.

## 2. Materials and Methods

### 2.1. Animals

All animal research protocols were approved by the Institutional Animal Care and Use Committee of Kaohsiung Veterans General Hospital (Identification code: vghks-2016-A007, vghks-2016-2019-A007, vghks-2017-A006, and vghks-2019-A005; date of approval: 11 June 2015, 19 May 2016, 19 May 2016, and 11 June 2018). Male Sprague–Dawley rats were obtained from BioLASCO Taiwan Co., Ltd. (Taipei, Taiwan). These rats were housed in the animal rooms of Kaohsiung Veterans General Hospital (Kaohsiung, Taiwan) and catheterized when their body weight reached 350–400 g.

### 2.2. E and/or NE Injection

Rats were divided into 4 groups: (1) sham group (injected with 0.9% NaCl) (Y F CHEMICAL CORP., New Taipei, Taiwan), (2) E group (injected with E) (TAIWAN BIOTECH Co., Ltd., Taoyuan, Taiwan), (3) NE group (injected with NE) (TAI YU CHEMICAL & PHARMACEUTICAL Co., Ltd., Hsinchu, Taiwan), and (4) E+NE group (injected with E and NE simultaneously). After anesthetizing the rats with urethane (1 g/kg, intraperitoneal; Sigma-Aldrich Co., St. Louis, MO, USA), their fur was shaved, and muscles and connective tissue were separated with hemostats, to expose the femoral vein. One end of the catheter was inserted into the femoral vein and the other end was connected with a syringe pump. The experimental groups were continuously injected with E (4.5 µg/kg/min) and/or NE (6.8 µg/kg/min) at a steady rate for 6 h during the experiment.

### 2.3. Ventricular Catheter Preparation

We connected one end of the PE-10 tubing (Becton, Dickinson and Company, Sparks, MD, USA) to one end of the PE-50 tubing (Becton, Dickinson and Company, Sparks, MD, USA) and fixed it with glue. The PE-50 tube was connected to the physiological pressure transducer via a three-way valve and filled with 10 IU heparin/0.9% saline (China Chemical & Pharmaceutical Co., Ltd., Taipei, Taiwan). The LV catheter tips of PE-10 tubing were straight and the RV catheter tips were curved [15,16].

### 2.4. Detecting Rat Heart Rate and Ventricular Pressure

After shaving the anesthetized rats’ neck fur, a section of the right jugular vein was separated from connective tissues, and the catheter was slowly moved forward and rotated into the RV. Similarly, the LV catheter also entered the LV via the left internal carotid artery. During the experiment, the rats’ heart rates and ventricular pressures were continuously recorded using a PowerLab data acquisition system and LabChart software (ADInstruments Inc., Colorado Springs, CO, USA) [17,18].

### 2.5. Catecholamine Levels

Blood samples from the rats’ hearts were collected in MiniCollect tubes (Greiner Bio-One GmbH, Kremsmünster, Austria) after 6 h of continuous catecholamine or 0.9% saline infusion. The concentration of E and NE was assayed using the enzyme-linked immunosorbent assay (ELISA) (2-CAT ELISA kit, #BA E-5400, Labor Diagnostika Nord, GmbH and Co.KG, Nordhorn, Germany) [10].

### 2.6. Histology and Quantitative Analysis of Immunohistochemistry

After the rats were sacrificed, the lungs and hearts were removed immediately. They were rinsed with iced 0.9% saline, cleared of blood and connective tissues, gently blotted with filter paper, and then weighed. Lung and heart tissue were placed in 10% formalin for 5 days, then embedded in paraffin for immunostaining.

The sections were immunostained by applying the Novolink Polymer Detection Systems (Leica Biosystems Newcastle Ltd., Newcastle Upon Tyne, UK) and incubated with diluted rabbit primary antibody, and then with anti-rabbit Poly-HRP-IgG reagent (Novolink Polymer) to recognize tissue-bound rabbit primary antibodies [19]. Finally, the sections were counterstained with hematoxylin and a coverslip was put in place. Results were observed using a BX51P polarizing microscope (OLYMPUS Corp., Center Valley, PA, USA).

We used the following rabbit polyclonal primary antibodies: anti-connexin43 (Cx43)/GJA1 antibody (diluted 1:200 for heart and 1:400 for lungs, Abcam, #ab11370, Cambridge, UK), anti-receptor for advanced glycation end products (RAGE) antibody (diluted 1:200 for lungs, GeneTex Inc., #GTX23611, Irvine, CA, USA), and anti-TNNT2 (cardiac) antibody (diluted 1:100 for heart, Cloud-Clone Corp., #PAD232Ra01, Houston, TX, USA). To assess the extent of Cx43 lateralization and troponin T [10] in cardiac tissue, we randomly selected three fields (magnification: 10 × 40) and counted the ratio of the number of Cx43-positive and Troponin T-positive cells in all cardiac cells in the RV, septum, and LV lateral wall areas of each rat heart section. The ratio of the number of Cx43-positive cells [9,10,20] and RAGE in all pulmonary cells in lung tissues [21] were calculated in three randomly selected fields of view (magnification: 10 × 40) selected from each lung section.

### 2.7. Echocardiography

As described in our previous studies [9,10], M-mode tracings and recordings were obtained from the parasternal short axis view for calculating interventricular septum thickness at end-diastole, LV internal dimension at end-diastole and end-systole, and posterior wall thickness at end-diastole, end-diastolic and end-systole volume, CO, ejection fraction (EF), fractional shortening, and LV mass. A pulsed-wave Doppler sample volume was positioned across the tricuspid and mitral valves, from the four-chamber view, to evaluate the diastolic function with early diastolic peak E wave and atrial filling peak A wave. Aortic flow was measured from the ascending aorta with the suprasternal view.

### 2.8. Statistical Analysis

All data are represented as mean ± SD. Data were analyzed using a nonparametric Kruskal–Wallis test followed by a Mann–Whitney U-test and a Wilcoxon signed rank test. A *p* value < 0.05 was considered statistically significant. All statistical analyses were performed using IBM SPSS Statistics Version 20 software (IBM Corp., Armonk, NY, USA, 2011).

## 3. Results

### 3.1. Overdose of NE and/or E Affected Heart Rate and Survival

To measure whether E or NE concentration was significantly elevated after infusion with 0.9% normal saline (sham), E, NE, and E+NE in respective groups for 6 h, the rats were sacrificed and the concentration of E and NE in plasma was measured (Figure 1a). The plasma E concentrations in the E and E+NE groups were significantly increased compared to those in the sham and NE groups (*p* < 0.05). The plasma NE concentration in the NE and E+NE groups was significantly raised compared to the other two groups (*p* < 0.05).

To examine whether overdose of E or NE would cause cardiomyopathy, the heart weight to body weight ratio was calculated, and showed a significant increase in the E, NE, and E+NE groups compared to that of the sham group after 6 h infusion (*p* < 0.05, Figure 1b). During the 6 h catecholamine injection, differences in heart rate between groups were compared at different time points (0, 1, 3, and 6 h time points, Figure 1c). The heart rate in the three treatment groups was significantly higher than that in the sham group after 1, 3, and 6 h of catecholamine injection (*p* < 0.05). In all the experimental groups, heart rate increased rapidly in the first hour of infusion, and reached a plateau at approximately the 3 h time point. At 6 h, only the NE group maintained a heart rate similar to the 3 h time point, while the E and E+NE groups showed a mild decrease in heart rate.

To understand whether the cardiomyopathy and change in heart rate may have influenced rat’s survival, hourly survival rates were also recorded during the infusion time (Figure 1d). After infusion of NE in the NE and E+NE group, lower survival rates were observed, with the E+NE group showing the lowest rat survival.

### 3.2. Overdose of NE Impaired Systolic and Diastolic Function More than E

To determine whether overdose of E and NE would influence the rats’ left ventricular systolic and diastolic function, echocardiography was used to record physiological heart function. Using M-mode images (Figure 2a), we found that left ventricular internal dimension in systole and diastole markedly declined in the NE and E+NE group but not in the E group after 6 h infusion (Appendix A).

According to echocardiography results, CO, stroke volume (SV), and EF could be calculated for systolic function (Figure 2b), while peak early diastolic mitral valve velocity (MV E Vel) and peak early/late diastolic mitral valve velocity could be presented as diastolic function (Figure 2c). We found that the NE group showed significantly reduced CO and SV compared to the sham group (*p* < 0.05). The E group showed only decreased SV compared to the sham group (*p* < 0.05) but the two groups were not different for CO. The combination of E and NE impaired systolic function, including SV and CO, more significantly than the E group (*p* < 0.05, Figure 2b). For diastolic function, a significant reduction in peak early/late diastolic mitral valve velocity and MV E Vel were found in the E, NE, and E+NE groups compared to the sham group at 6 h (*p* < 0.05). Lower MV E Vel was observed in the NE and E+NE groups than in the E group (Figure 2c). Additionally, we observed that interventricular septum thickness, left ventricular posterior wall thickness, and the total left ventricular mass, significantly increased in the NE and E+NE groups when compared to the sham group (Appendix A).

### 3.3. Overdose of NE and/or E Led to Presence of Cardiac Damage Markers

Cardiac troponin release has been measured to diagnose different types of acute cardiac injuries while the lateralization of Cx43 can be observed in many pathological heart diseases [22,23,24]. Immunohistochemical staining was performed to detect troponin T and Cx43 in cardiac tissue after 6 h infusion of catecholamine (Figure 3a,b). Our results showed that the expression of troponin T and Cx43 was significantly increased in the E and NE groups and the E+NE group showed greater changes in the RV, septum, and LV lateral walls compared to the sham group. In these three parts, lateralization of Cx43 was also raised in the NE and E+NE groups and the damage appeared in whole ventricular cardiac muscles rather than specific regions (Figure 3a).

### 3.4. Overdose of NE rather than E Led to Pulmonary Edema and Damage

Because of the acute injury of cardiac muscle sustained from catecholamine overdose, we also assessed whether it would cause severe pulmonary edema and damage. After a 6 h catecholamine injection, the lung tissue was collected, and hematoxylin-and-eosin staining was performed to observe histological changes (Figure 4a). A thickened alveolar wall was observed in the E, NE, and E+NE groups compared to the sham group. However, interstitial edema was found only in the NE and E+NE groups. The NE and E+NE groups showed a significant increase in their lung weight to body weight ratio compared to the other two groups (Figure 4b). Given that previous studies reported that RAGE downregulation and Cx43 upregulation may play potential roles in acute lung injuries [25,26], we assessed changes in the expression of RAGE and Cx43 in the lung tissue after 6 h continuous catecholamine infusion, via immunohistochemical staining (Figure 4c). In the E, NE, and E+NE groups, the ratio of RAGE-positive cells in all pulmonary cells significantly decreased (*p* < 0.05) and the ratio of Cx43-positive cells in total pulmonary cells significantly increased (*p* < 0.05) compared to the sham group (Figure 4d).

### 3.5. RV Was More Sensitive and More Vulnerable to Toxicity of Catecholamine Overdose

After observing the occurrence of lung injuries in the catecholamine-treated groups (E, NE, and E+NE), we hypothesized that these injuries were due to the high pressure of the RV. To understand whether there were different pressure patterns between the LV and RV, we also recorded pressure changes in both ventricles during catecholamine infusion. Table 1 demonstrated LV and RV pressure at different time points, while Figure 5a,b showed the percentage increase compared to 0 h results. In the sham group, the LV systolic pressure (SP) increased at the 3 h and 6 h time points compared to the 0 h and 1 h time points due to normal saline volume expansion. However, the LV end-diastolic pressure (EDP) did not change significantly during the injection period. For treatment groups, left ventricular systolic pressure (LVSP) significantly increased at 1, 3, and 6 h time points compared to the 0 h in the E, and NE groups. Moreover, LVSP significantly increased at 1, and 3 h time points compared to the 0 h in the E+NE groups. However, LVSP and left ventricular end-diastolic pressure (LVEDP) in the E+NE group markedly decreased at the 6 h time point when they were compared to the results of 1 and 3 h time points due to decompensated heart failure. Regarding the changes of RV pressure, in the sham group, right ventricular systolic pressure (RVSP) and right ventricular end-diastolic pressure (RVEDP) did not change significantly during the 6 h period of infusion. In the E, NE, and E+NE groups, RVSP and RVEDP were significantly elevated after 1 h when compared to the sham group, and RVSP remained elevated after 3 h. However, both RVSP and RVEDP in the NE and E+NE groups had the highest value at the 1 h time point, and the elevation level of these two pressures still declined in a time-dependent manner.

For increased percentage change of ventricular pressure (Figure 5a,b), after 1 h of continuous catecholamine infusion, we found that the systolic and diastolic pressure change is greater in the RV (increased 5.1% and 23% in the sham group, 61.8% and 95.8% in the E group, 68.5% and 164.6% in the NE group, and 64.6% and 118.6% in the E+NE group) than in the LV (increased 1.8% and 4.3% in the sham group, 36.1% and 77.2% in the E group, 35.3% and 46.3% in the NE group, and 37.0% and 93.5% in the E+NE group) under high plasma levels of E and/or NE.

## 4. Discussion

Using a rat model of excessive catecholamine administration, we found a significant increase in both E and NE serum concentrations, which led to acute tachycardia and demonstrated the efficacy of the catecholamine injections. The cardiac pathology results showed that high levels of catecholamines caused diffuse acute myocardial damage in the region of the ventricles. However, the damage induced by catecholamines may not be in proportion to the level of cardiac dysfunction. The echocardiography results demonstrated that LV systolic and diastolic dysfunction were more affected by NE than E during the 6 h consecutive administration. In addition to LV dysfunction, RV systolic and diastolic dysfunctions were also recorded, and these effects were more severe and rapid than in the LV during the 6 h testing. RV dysfunctions may also cause lung injuries; yet we found that NE induced more pulmonary edema and damage than E.

From the 2019 ACLS guideline update, there were no survival benefits of high-dose E compared to standard-dose E [2]. Although few studies with large sample sizes have directly reported the toxicity of catecholamines, a number of case reports have documented the link between E overdose with a series of arrhythmia changes including sinus tachycardia, and idioventricular rhythm with multifocal premature ventricular complex [27] or ST elevation due to myocardial injury [28]. Another review proposed four plausible mechanisms of catecholamine-induced cardiotoxicity [29]. All these studies indicate that catecholamine administration beyond the maximal doses may lead to cardiac dysfunction; therefore, it is important to investigate the cardiopulmonary damages induced by excessive E and NE and their potential mechanisms.

Previously, some animal models were designed to simulate catecholamine overdose, even though they used different doses and administration routes [30,31]. In our study, we chose the maximal doses of E and NE adequate for inducing significant cardiopulmonary damages without extensively affecting survival during our observational time interval. One study reported that E leads to LV apical hypokinesis [31]; however, in our study the different parts of the LV were not analyzed; but the results showed that NE overdose impaired SV, CO, and MV E Vel more significantly than E, which may suggest a greater risk of cardiac systolic and diastolic dysfunction in the NE group (Figure 2). In addition, the similarities of the results between the NE and E+NE groups, may indicate a predominant role of NE in determining the overall cardiopulmonary damage.

Though few studies have investigated the mechanisms of catecholamine overdose thoroughly, the pathophysiological hypothesis of the Takotsubo cardiomyopathy may provide a valid mechanistic model as this disease is also provoked by high levels of catecholamines [32]. One review has described a phenomenon called “stimulus trafficking” where E stimulates the negative inotropic effects through shifting β2 AR from Gs- to Gi-coupling while NE keeps stimulating β1 AR-Gs coupling [33]. With this shift, supra-physiological concentration of E-induced cardiotoxicity can be alleviated, which may explain the cardiac functional differences observed in our study (Figure 1 and Figure 2).

However, cardiac pathology showed no differences between E- and NE-induced myocardial damage, which may indicate catecholamine-mediated toxicity independent of stimulus trafficking. NE-induced apoptosis of a rat cardiomyocyte may be selectively mediated by β1 AR rather than β2 AR [34]. Yet, no analysis of E was conducted in the same study. We also performed a Terminal deoxynucleotidyl transferase dUTP nick end labeling assay to assess whether E+NE would induce more apoptosis compared to the sham group, but no significant differences were found (Appendix A); therefore, apoptosis may not play an important role in catecholamine overload. A variety of cellular responses may be induced by catecholamine overload, including extracellular collagen accumulation, increased reactive oxidative stress (ROS), and activating transforming growth factor β [35]. One study shows that elevated plasma E levels induce ROS in cardiomyocytes leading to troponin release [36]. ROS also reduces gap junction coupling and increases Cx43 lateralization [37]. Therefore, we hypothesized that these two catecholamines may stimulate massive ROS accumulation in myocardium and cause further myocardial injury. This hypothesis should be further investigated in future studies.

A central nervous system disorder (i.e., neurogenic pulmonary edema (NPE)) may also mimic catecholamine overload. The destruction of the medulla oblongata, especially NTS, is involved in the pathogenesis of NPE. It is thought that brain lesions determine sympathetic overactivation and a large release of catecholamines may increase pulmonary capillary hydrostatic pressure [38] and the onset of pulmonary edema. A study showed that patients with subarachnoid hemorrhage and NPE had a higher concentration of NE rather than E, when they were compared to patients without NPE [39]. Another article demonstrated that NE-induced NPE is more α-AR-dependent compared to β suggesting that α-adrenergic antagonists may have better attenuation effects [40]. These results seem to suggest that overdose of NE may be more closely associated with the onset of pulmonary edema compared to E, and the mechanism appears to be α-AR-dependent.

RAGE is expressed primarily in alveolar type I cells [41]. When acute lung injury occurs, RAGE will be released into pulmonary interstitium, therefore downregulating the expression of RAGE in alveolar cells [25]. In addition, Cx43 will be upregulated when lung injuries such as radiation-induced fibrosis occur [26]. We have shown that 6-OHDA-induced NTS lesions may cause pulmonary damage through elevating serum catecholamine levels in rats [10]. Furthermore, this study showed that overdose of E, NE, and E+NE with a 6 h infusion reduced RAGE-positive cell percentage while Cx43-positive cell percentage increased. These results suggest that both E and NE lead to acute pulmonary damage, which is in line with our previous findings.

Prior to our experiment, Rassler et al. used an animal model to compare RV and LV systolic function in order to understand catecholamine-induced (0.1 mg/kg/h NE) differences in responses between the two ventricles. This study indicated that NE increases maximal RVSP and LVSP significantly within 6 to 24 h, and RVSP is elevated more dramatically than LVSP [40]. In our study, we also found a more apparent and rapid rise in SP and EDP of the RV than those of the LV at a very early stage, indicating the RV’s greater vulnerability to catecholamines. Since, to the best of our knowledge, there are no reports of this mechanism, we speculate some putative mechanisms accounting for these cardiac events. First, both the myocardial wall and the volume of the LV is thicker and larger, respectively, than the RV, making the LV more resistant to pressure change. Moreover, the density of ARs are different between the RV and LV and thus, both ventricles have distinct responses to catecholamines [42].

There are some limitations in this study. Firstly, only animal models were used in this research, and results may not be representative of human physiology. No AR analysis was conducted in the research and, although the effects of NE on cardiac and pulmonary functions during continuous IV treatment are known, it remains to be established which receptors are directly related to the onset of the disorders. Future studies should investigate whether the more harmful effects of NE are due to different a distribution of AR on the heart and/or different AR affinities between NE and E. Secondly, we did not investigate the cardiac pro-inflammatory effects for the pathways of NF-κB and interleukins in the pathogenesis of cardiac injury [43]. An additional limitation of the study was that we could not determine whether the pulmonary damage was cardiac-dependent or independent. Finally, only healthy rats were included in this research and it is uncertain whether there would be more harm than benefit when NE or E is treated during severe conditions such as septic shock or acute cardiac arrest.

## 5. Conclusions

In summary, overdose of NE may lead to more significant cardiopulmonary impairment and dysfunction compared to E. Even though only animal models were used to simulate catecholamine overload in the current study, and no other treatment was administered to reverse the disorders, this study still provides a warning about the excessive exogenous exposure to catecholamines. At the same time, the RV may be more vulnerable to catecholamine overload than the LV, which has received limited attention in previous studies investigating excessive catecholamine treatment models. The more rapid and higher increase in RV pressure induced by catecholamine administration, may explain the occurrence of acute lung injuries and pulmonary edema. We aim to determine the deterioration mechanism as consequence of catecholamine overload in future work.

## Figures and Tables

**Figure 1 toxics-08-00069-f001:**
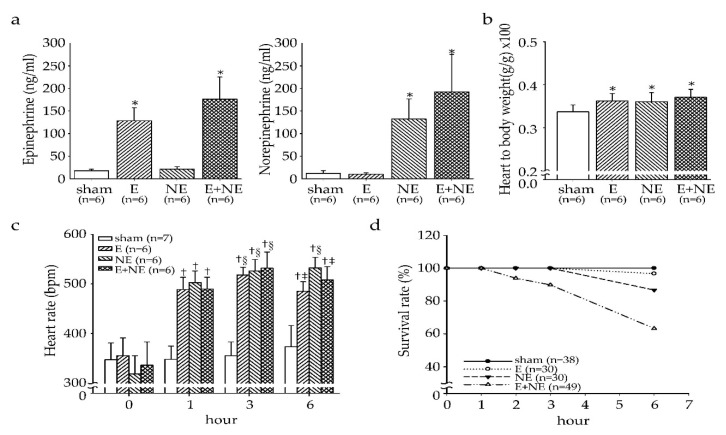
The effect of catecholamine overdose on heart rate and survival rate. (**a**) The concentration of epinephrine and norepinephrine in 4 groups. (**b**) The ratio of heart to body weight in the 4 groups. (**c**) Heart rate was detected during 6 h injection of catecholamine. (**d**) The percentage of survival rate after continuous catecholamine infusion. Data are expressed as means ± SD. Sham, 0.9% saline injection group; E, epinephrine injection group; NE, norepinephrine injection group; E+NE, epinephrine and norepinephrine injection group. * *p* < 0.05 vs. sham group. ^†^
*p* < 0.05 vs. respective sham group and *p* < 0.05 vs. respective group at 0 h. ^§^
*p* < 0.05 vs. respective group at 1 h. ^‡^
*p* < 0.05 vs. respective group at 3 h.

**Figure 2 toxics-08-00069-f002:**
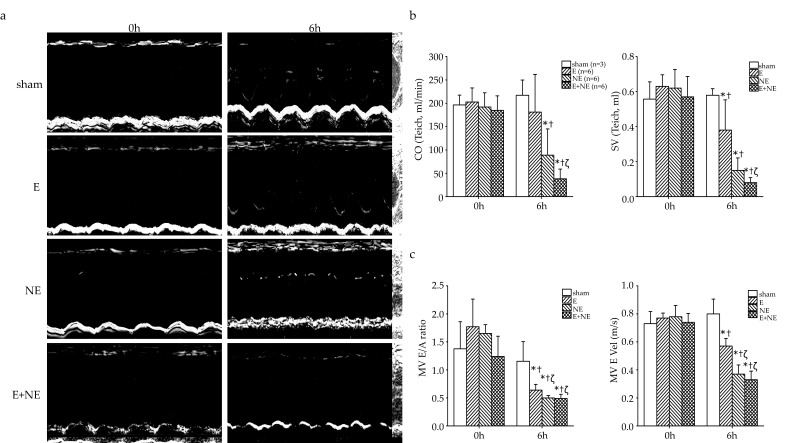
The cardiac functional change induced by excessive catecholamine infusion. (**a**) Representative M-mode echocardiograms of rats infused continuously catecholamine at 0 and 6 h. Bar graphs were shown for echocardiographic quantitative data: (**b**) CO, SV, and (**c**) MV E/A ratio and MV E Vel. Data are expressed as means ± SD. Sham, 0.9% saline; E, epinephrine; NE, norepinephrine; E+NE, epinephrine and norepinephrine; CO, cardiac output; SV, Stroke volume; MV E/A ratio, peak early/late diastolic mitral valve velocity; MV E vel, peak early diastolic mitral valve velocity. * *p* < 0.05 vs. sham group at 6 h, ^†^
*p* < 0.05 vs. respective group at 0 h, ^ζ^
*p* < 0.05 vs. Epinephrine group.

**Figure 3 toxics-08-00069-f003:**
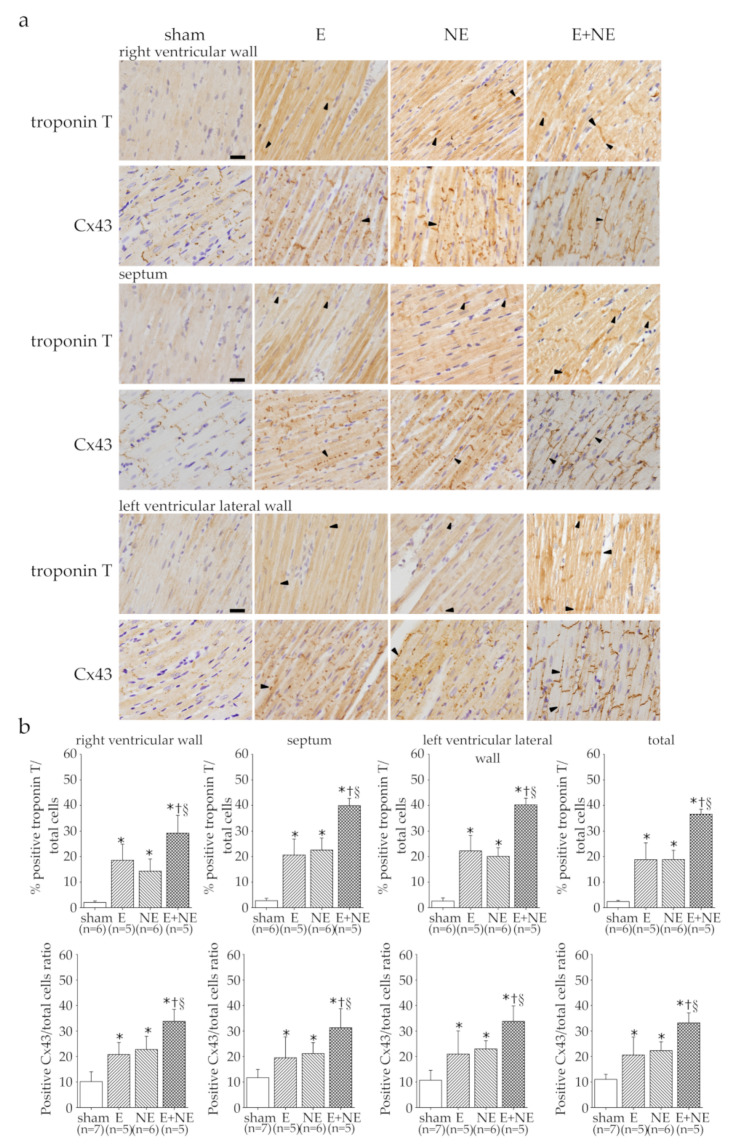
The myocardial injuries induced by catecholamine overdose. (**a**) Representative examples of troponin T and lateralized connexin43 (arrowhead) in heart sections of sham, E, NE and E/NE combined treatment groups. (**b**) Bar graphs showing quantitative analysis of troponin T and the lateralized population of connexin43 in the right ventricle, septum, and left ventricular lateral wall of the rat heart. Data are expressed as means ± SD. Cx43, connexin43; sham, 0.9% saline; E, epinephrine; E+NE, epinephrine and norepinephrine; NE, norepinephrine. Scale bar = 50 μm. * *p* < 0.05 vs. sham group. ^†^
*p* < 0.05 vs. E group. ^§^
*p* < 0.05 vs. NE group.

**Figure 4 toxics-08-00069-f004:**
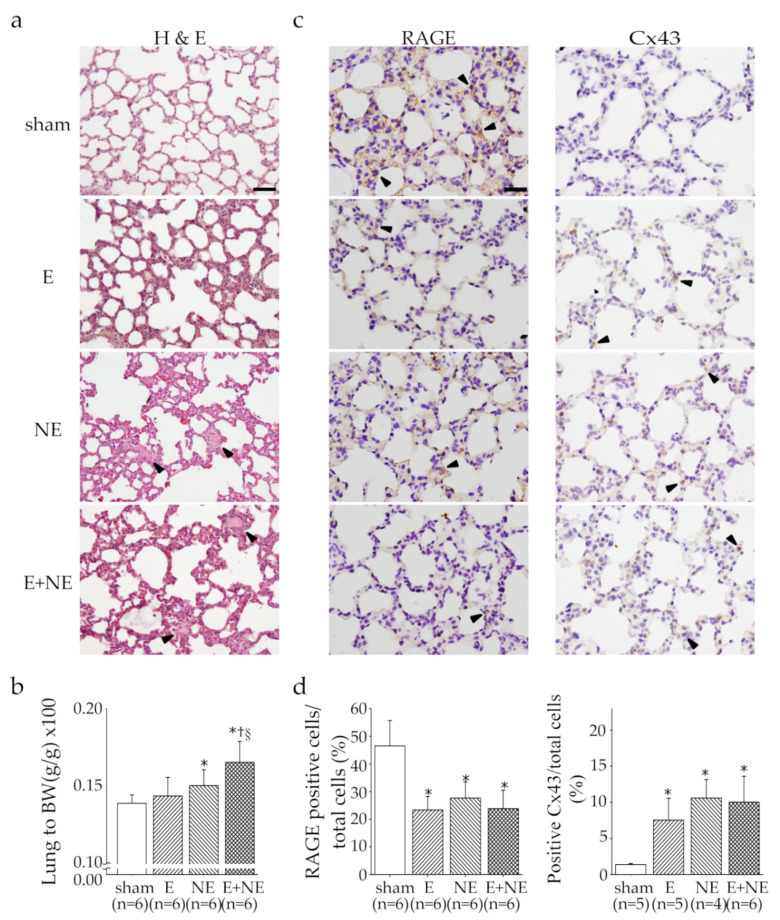
The acute lung injuries induced by catecholamine overdose. (**a**) Representative hematoxylin-and-eosin (H and E) staining of lung sections after 6 h continuous catecholamine injection and (**b**) the ratio of lung to body weight was showed. (**c**) Representative immunohistochemical staining with RAGE and connexin43 antibody (arrowhead) in lung sections of sham, E, NE and E+NE groups. (**d**) Quantitative analysis of RAGE and connexin43 in lung tissues. Data are expressed as means ± SD. RAGE, the receptor for advanced glycation end products; Cx43, connexin43; sham, 0.9% saline; E, epinephrine; NE, norepinephrine; E+NE, epinephrine and norepinephrine. Scale bar = 50 μm. * *p* < 0.05 vs. sham group, ^†^
*p* < vs. E group, ^§^
*p* < 0.05 vs. NE group.

**Figure 5 toxics-08-00069-f005:**
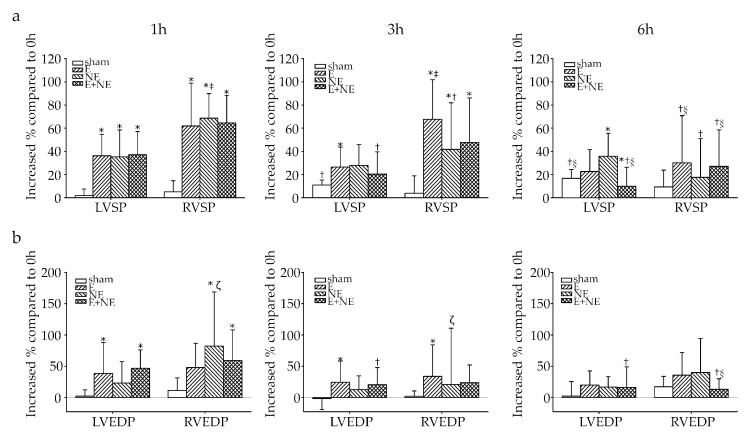
The comparison between left and right ventricular pressure change during 6 h intravenous continuous catecholamine administration. (**a**) The increased percentage of ventricular systolic pressure compared to respective group at 0 h. (**b**) The increased percentage of ventricular diastolic pressure compared to respective group at 0 h. Data are expressed as means ± SD. LV, left ventricle; RV, right ventricle; SP, systolic pressure; EDP, end- diastolic pressure; sham, 0.9% saline; E, epinephrine; NE, norepinephrine; E+NE, epinephrine and norepinephrine. * *p* < 0.05 vs. sham, ^†^
*p* < 0.05 vs. respective group at 1 h, ^§^
*p* < 0.05 vs. respective group at 3 h, ^‡^
*p* < 0.05 vs. LVSP of respective group, ^ζ^
*p* < 0.05 vs. LVEDP of respective group.

**Table 1 toxics-08-00069-t001:** Left and right ventricular pressure of rats during intravenous continuous administration of epinephrine and/or norepinephrine.

Parameters	Ventricular Pressure (mmHg)				
	Hour	Sham (*n* = 7)	E (*n* = 7)	NE (*n* = 7)	E+NE (*n* = 8)
LVSP	0 h	86.2 ± 11.1	90.3 ± 9.7	89.0 ± 11.1	87.0 ± 8.7
	1 h	87.8 ± 13.0	122.4 ± 18.7 *^,†^	118.4 ± 8.2 *^,†^	118.4 ± 16.4 *^,†^
	3 h	95.2 ± 10.0 ^†,‡^	114.0 ± 20.3 *^,†^	112.4 ± 11.5 *^,†^	104.4 ± 16.9 ^†,‡^
	6 h	100.5 ± 14.21^†,‡,§^	111.1 ± 24.4 ^†^	119.3 ± 17.7 ^†^	95.1 ± 12.9 ^‡,§,^^¥^
LVEDP	0 h	11.9 ± 5.1	9.0 ± 2.8	11.8 ± 6.0	11.0 ± 3.9
	1 h	12.3 ± 5.5	16.7 ± 12.5 ^†^	19.9 ± 15.1	20.8 ± 8.3 *^,†^
	3 h	11.2 ± 5.6	14.4 ± 9.5	16.7 ± 11.8	15.2 ± 7.2 ^‡^
	6 h	12.3 ± 7.1	13.1 ± 7.2	17.2 ± 11.3	13.3 ± 5.2 ^‡^
RVSP	0 h	17.2 ± 3.0	17.3 ± 3.6	16.8 ± 2.5	18.4 ± 3.9
	1 h	18.2 ± 4.2	27.3 ± 5.1 *^,†^	27.8 ± 2.1 *^,†^	29.8 ± 5.6 *^,†^
	3 h	17.9 ± 4.1	28.3 ± 5.1 *^,†^	22.9 ± 2.9 *^,†,‡,#^	26.4 ± 6.0 *^,†^
	6 h	18.8 ± 4.0	21.6 ± 5.3 ^‡,§^	19.3 ± 4.9 ^‡^	23.0 ± 6.2 ^†,‡,§^
RVEDP	0 h	7.3 ± 3.1	8.0 ± 1.6	9.0 ± 4.1	8.1 ± 3.0
	1 h	8.2 ± 3.4	15.4 ± 5.2 *^,†^	18.9 ± 4.0 *^,†^	15.5 ± 4.5 *^,†^
	3 h	8.2 ± 3.5	17.0 ± 5.8 *^,†^	15.1 ± 3.1 *^,†^	13.2 ± 5.6 *^,†^
	6 h	9.2 ± 3.9 ^†^	13.2 ± 4.5 ^†,§^	12.4 ± 4.1 ^‡^	10.2 ± 4.2 ^‡,§^

Data are expressed as means ± SD. E, epinephrine group; E+NE, epinephrine and norepinephrine group; LVEDP left ventricular end-diastolic pressure; LVSP, left ventricular systolic pressure; NE, norepinephrine group; RVEDP, right ventricular end-diastolic pressure; RVSP, right ventricular systolic pressure; sham, 0.9% normal saline group. * *p* < 0.05 vs. sham group. ^†^
*p* < 0.05 vs. respective group at 0 h. ^‡^
*p* < 0.05 vs. respective group at 1 h. ^§^
*p* < 0.05 vs. respective group at 3 h. ^#^
*p* < 0.05 vs. E group. ^¥^
*p* < 0.05 vs. NE group.

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
