# Peer review of "Norepinephrine Leads to More Cardiopulmonary Toxicities than Epinephrine by Catecholamine Overdose in Rats"

_toxics, 2020, doi:10.3390/toxics8030069_

Round 1

Reviewer 1 Report

Good paper. I would like to suggest you to analyse the cytokines involved in noreprinephrine and epinephrine-related cardiovascular effects. A partucular attention should be made on the role of IL1 IL6 IL8 in cardiac function (you can cite "Quagliariello V, Passariello M, Maurea N, et al.,Int J Cardiol. 2019)

After this modification the manuscript should be acceptable for publication.

Reviewer 2 Report

The manuscript of Lu and coworkers focuses on the cardiopulmonary toxicities of norepinephrine and epinephrine in rats. They found a more severe cardiopulmonary impairment after over dose with norepinephrine compared to epinephrine. The manuscript is in principle well written, but some sentences are very long and should be shortened or split for better understanding by the reader.

Abstract:

Line 31: “greater dysfunction” à change wording

Introduction:

Line 44: Remove activities à The autonomic nervous system regulates…

Line 47-50: Please split sentence and check content afterwards carefully

Line 67: “The tumor mainly secrets NE” à This is not completely correct, because there are predominantly adrenergic or noradrenergic pheochromocytomas. Read the common literature and correct. Furthermore, signs and symptoms differ between pheochromocytomas with a noradrenergic and an adrenergic phenotype.

Materials and Methods:

Line 104: Rats were under narcosis for the entire period of the injection?

Line 108: For injection of E+NE the same dosages as for the single treatments where used? Injection rate?

Line 138-142: Add catalog number of the primers

Results:

Line 178-179: “a rat´s surivial” à remove “a”

Line 192-193: Remove the first sentence of this section. Doubling with the introduction and not required for the understanding of this section

Figure 2b: CO (Teich, ml/min) and SV (Teich, ml/min). What does “Teich” mean? In the figure legend also EF data are mentioned, but they are not shown

Line 268: noral à normal

Discussion:

Line 320: What is the maximal dose? And in what dose range do we compare in this study?

Line 330-331: This is really interesting. I would add this also to the abstract

Line 361-362: See comment above
